# Critical spin liquid versus valence-bond glass in a triangular-lattice organic antiferromagnet

Kira Riedl[1], Roser Valentí[1] & Stephen M. Winter [1]

In the quest for materials with unconventional quantum phases, the organic triangular-lattice antiferromagnet $\kappa$-(ET)$_2$Cu$_2$(CN)$_3$ has been extensively discussed as a quantum spin liquid (QSL) candidate. The description of its low temperature properties has become, however, a particularly challenging task. Recently, an intriguing quantum critical behaviour was suggested from low-temperature magnetic torque experiments. Here we highlight significant deviations of the experimental observations from a quantum critical scenario by performing a microscopic analysis of all anisotropic contributions, including Dzyaloshinskii–Moriya and multi-spin scalar chiral interactions. Instead, we show that disorder-induced spin defects provide a comprehensive explanation of the low-temperature properties. These spins are attributed to valence bond defects that emerge spontaneously as the QSL enters a valence-bond glass phase at low temperature. This theoretical treatment is applicable to a general class of frustrated magnetic systems and has important implications for the interpretation of magnetic torque, nuclear magnetic resonance, thermal transport and thermodynamic experiments.

[1] Institut für Theoretische Physik, Goethe-Universität Frankfurt, Max-von-Laue-Strasse 1, 60438 Frankfurt am Main, Germany. Correspondence and requests for materials should be addressed to K.R. (email: riedl@th.physik.uni-frankfurt.de) or to S.M.W. (email: winter@physik.uni-frankfurt.de)

Despite intensive efforts devoted to uncover the nature of the low-temperature properties of the quantum spin liquid (QSL) candidate $\kappa$-(ET)$_2$Cu$_2$(CN)$_3$ (ET = bisethyelenedithioltetrathiafulvalene)[1–3], a few puzzles remain unresolved, such as the presence of an anomaly near a temperature $T^* = 6$ K in a wide variety of experiments, including $^{13}$C nuclear magnetic resonance (NMR)[4], muon spin resonance ($\mu$SR)[5,6], electron spin resonance (ESR)[7], specific heat[8,9], ultrasound attenuation[10], and thermal expansion[9]. Various scenarios have been suggested to explain this anomaly such as spin-chirality ordering[11], a spinon-pairing transition[12–15], or formation of an exciton condensate[16]. However, a comprehensive explanation of all aspects of the anomaly and its possible relation to a spin liquid phase is currently lacking.

Motivated by the unconventional scaling observed in magnetic torque measurements, Isono et al.[17] suggested a low-temperature quantum critical scenario for $\kappa$-(ET)$_2$Cu$_2$(CN)$_3$ ($\kappa$-Cu). A finite torque magnitude $\tau$ reflects a variation of the energy $E$ as a function of the orientation of the external magnetic field **H** (Fig. 1a)

$$\tau = \frac{dE}{d\theta}, \tag{1}$$

where $\theta$ is the angle between **H** and a reference axis. Conventionally, $\tau$ is considered to probe the uniform magnetization **M**, through $\tau \propto \mathbf{M} \times \mathbf{H}$. However, the torque is sensitive to any angular variation of the energy, which allows more general instabilities to be probed.

The experimentally observed torque response of $\kappa$-Cu[17] displays several characteristic features. As **H** is rotated in the crystallographic $ac^*$-plane, $\tau$ follows a sinusoidal angle dependence $\tau \propto \sin 2(\theta - \theta_0)$, with an angle shift $\theta_0$ that increases substantially at low $T$ and $H$, suggesting the emergence of a new contribution below $T^*$. Indeed, the torque susceptibility $\chi_\tau = \tau/H^2$ diverges as a power law with the temperature exponent $\omega$ at low fields, $\chi_\tau \sim T^{-\omega}$, and with the field exponent $\zeta$ at low temperatures, $\chi_\tau \sim H^{-\zeta}$. In experiment, these exponents are approximately the same[17] $\omega \approx \zeta \approx 0.8$. Consequently, the torque displays an apparent critical $H/T$ scaling over several orders of magnitude. In this context, the remarkable behavior of $\kappa$-Cu has been interpreted[17] in terms of a field-induced quantum critical point with a diverging uniform spin susceptibility.

However, the presence of magnetic interactions beyond the conventional Heisenberg couplings allows more general instabilities of the spin liquid to be directly probed by torque measurements. For example, in $\kappa$-Cu, spin–orbit coupling (SOC) leads to a staggered $g$-tensor and a finite Dzyaloshinskii–Moriya (DM) interaction[18]. These terms ensure torque contributions from the staggered magnetic susceptibility, which may diverge near an instability toward a canted Néel antiferromagnetic order[19]. Furthermore, higher order multi-spin interactions[20] couple the field to the scalar spin chirality, allowing $\tau$ to probe transitions to chiral phases[11].

In the following we show that through a microscopic calculation of $\tau$, the quantum critical scenario cannot account for the experimental findings, even when all such instabilities are considered. Instead, motivated by the observation of an inhomogeneous NMR response below $T^*$, we consider disorder-induced effects[21,22]. In particular, we show that the NMR and $\tau$ experiments are consistent with proximity to a finite randomness large-spin fixed point (LSFP)[23,24], where the diverging $\chi_\tau$ reflects a small density of quasi-free defect spins. By considering the specific temperature dependence of the response, the anomaly at the temperature $T^*$ can be interpreted as the onset of a valence bond glass (VBG), at which the resonating valence bonds of the QSL become randomly pinned. Finally, we discuss how this interpretation is consistent with a remarkable range of experiments.

## Results

**Effective Hamiltonian.** We first consider possible anisotropic terms in the spin Hamiltonian that contribute to $\tau$. These include anisotropic exchange interactions and $g$-tensor anisotropy that arise from SOC, as well as higher order spin-chiral terms. Ab initio estimates of each contribution are detailed in Supplementary Note 1.

The bilinear exchange interactions can be written in terms of the Heisenberg exchange $J_{ij}$, the Dzyaloshinskii–Moriya (DM) vector $\mathbf{D}_{ij}$, and the pseudo-dipolar tensor $\Gamma_{ij}$

$$\mathcal{H}_{ij} = J_{ij} \mathbf{S}_i \cdot \mathbf{S}_j + \mathbf{D}_{ij} \cdot (\mathbf{S}_i \times \mathbf{S}_j) + \mathbf{S}_i \cdot \Gamma_{ij} \cdot \mathbf{S}_j, \tag{2}$$

where $\mathbf{S}_i$ denotes the spin at site $i$. For $\kappa$-Cu, the sites consist of molecular dimers, which are arranged on an anisotropic triangular lattice with two dimer sublattices, labeled A and B in Fig. 1b. We have previously estimated the bilinear interactions including SOC for $\kappa$-Cu in ref. [18], and found that the DM-vectors are nearly parallel, but staggered ($\mathbf{D}_{ij} \approx \pm \mathbf{D}$), with magnitude $|\mathbf{D}|/J \sim 5\%$. It is therefore useful to transform the Hamiltonian[25] by applying local staggered rotations of the spins $\mathbf{S}_i \rightarrow \tilde{\mathbf{S}}_i$ around **D** by a canting angle $\pm \phi_i$. For a purely staggered **D**-vector, this eliminates the DM interaction and also, in leading orders, the pseudo-dipolar tensor. As a result, the transformed bilinear interactions are

$$\mathcal{H}_{ij} \approx \tilde{J}_{ij} \tilde{\mathbf{S}}_i \cdot \tilde{\mathbf{S}}_j, \tag{3}$$

with $\tilde{J}_{ij} \approx J_{ij}$ and $J_{ij}/k_B \sim 230$–270 K. Since the transformed bilinear interactions are rotationally invariant, they do not explicitly contribute to $\tau$. Instead, the anisotropic effects are

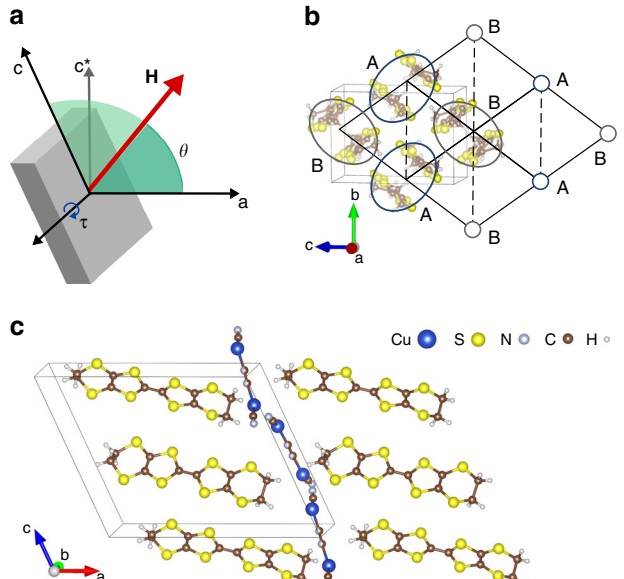

**Fig. 1** Magnetic torque in the organic charge transfer salt $\kappa$-Cu. **a** Definition of quantities used in the expressions of the torque $\tau$. $\theta$ is the angle between the magnetic field **H** and the principle axis of the product of the uniform $g$-tensors, $(\mathbb{G}_u \cdot \mathbb{G}_u^T)$, which is approximately the crystallographic $a$-axis. **b** Anisotropic triangular lattice with the sublattices labeled A and B on top of the bisethyelenedithioltetrathiafulvalene (ET) dimers in $\kappa$-(ET)$_2$Cu$_2$(CN)$_3$ ($\kappa$-Cu). **c** Crystal structure of $\kappa$-Cu, showing the organic layer of ET molecules and the anion layer

shifted to the transformed Zeeman term:

$$\mathcal{H}_{\mathrm{Zee,eff}} = -\mu_{\mathrm{B}} \mathbf{H} \cdot \sum_i \left( \tilde{\mathbb{G}}_{\mathrm{u}} + \eta_i \tilde{\mathbb{G}}_{\mathrm{s}} \right) \cdot \tilde{\mathbf{S}}_i, \tag{4}$$

where $\tilde{\mathbb{G}}_u$ and $\tilde{\mathbb{G}}_s$ are the uniform and staggered components of the g-tensor in the rotated frame, and $\eta_i = +1(-1)$ for $i \in \mathrm{A(B)}$ sublattice. Microscopic expressions for $\tilde{\mathbb{G}}_{\mathrm{u}}$ and $\tilde{\mathbb{G}}_{\mathrm{s}}$ are detailed in Supplementary Note 1.

For materials close to the Mott transition (such as κ-Cu), it has been suggested that higher order ring-exchange terms such as $\mathcal{H}_{(4)} = 1/S^2 \sum_{\langle ijkl \rangle} \tilde{K}_{ijkl} \left( \tilde{\mathbf{S}}_i \cdot \tilde{\mathbf{S}}_j \right) \left( \tilde{\mathbf{S}}_k \cdot \tilde{\mathbf{S}}_l \right)$, where $\tilde{K}_{ijkl}$ is the ring exchange parameter in the rotated coordinates, may also play a significant role in stabilizing QSL states[20,26,27]. As detailed in Supplementary Note 1, our ab initio estimates find $\tilde{K}/\tilde{J} \sim 0.1$, which represents a significant contribution[20,26]. Including SOC leads to anisotropic ring-exchange terms, in principle. Fortunately, the leading order contributions to these terms are also eliminated by the $\mathbf{S} \rightarrow \tilde{\mathbf{S}}$ transformation, such that the transformed 4-spin ring exchange terms do not explicitly contribute to $\tau$ at lowest order.

Finally, additional 3-spin scalar chiral interactions[20] may also arise when a finite magnetic flux penetrates the 2D organic layers. In the rotated coordinates, this provides the interaction

$$\mathcal{H}_{\Phi,\mathrm{eff}} = -\mu_{\mathrm{B}} j_\Phi (\mathbf{H} \cdot \mathbf{n}) \sum_{\langle ijk \rangle} \tilde{\mathbf{S}}_i \cdot \left( \tilde{\mathbf{S}}_j \times \tilde{\mathbf{S}}_k \right), \tag{5}$$

where $\mathbf{n}$ is the out-of-plane unit vector and the parameter $j_\Phi$ is proportional to the magnetic flux $\frac{\hbar}{q}\Phi$ enclosed by the triangular plaquette $\langle ijk \rangle$. Since only the out-of-plane component of the magnetic field couples to the scalar spin chirality, this term contributes explicitly to $\tau$. We estimate $\mu_{\mathrm{B}} j_\Phi / k_{\mathrm{B}} \sim 0.03 K/T$.

**Bulk torque expressions near a quantum critical point.** In the rotated frame, the bulk contribution to $\tau$ depends only on two terms in the Hamiltonian

$$\tau = \frac{\mathrm{d}\langle \mathcal{H}_{\mathrm{Zee,eff}} \rangle}{\mathrm{d}\theta} + \frac{\mathrm{d}\langle \mathcal{H}_{\Phi,\mathrm{eff}} \rangle}{\mathrm{d}\theta}. \tag{6}$$

For notational convenience, we introduce effective fields, $\mathbf{H}_{\mathrm{eff,u/s}} = \tilde{\mathbb{G}}_{\mathrm{u/s}}^{\mathrm{T}} \cdot \mathbf{H}$ and $\mathbf{H}_{\mathrm{eff,\Phi}} = j_\Phi (\mathbf{n} \cdot \mathbf{H}) \mathbf{n}$, which couple directly to the rotated spin variables

$$\mathcal{H}_{\mathrm{Zee,eff}} = -\mu_{\mathrm{B}} \sum_i \left( \mathbf{H}_{\mathrm{eff,u}} + \eta_i \mathbf{H}_{\mathrm{eff,s}} \right) \cdot \tilde{\mathbf{S}}_i, \tag{7}$$

$$\mathcal{H}_{\Phi,\mathrm{eff}} = -\mu_{\mathrm{B}} |\mathbf{H}_{\mathrm{eff,\Phi}}| \sum_{\langle ijk \rangle} \tilde{\mathbf{S}}_i \cdot \left( \tilde{\mathbf{S}}_j \times \tilde{\mathbf{S}}_k \right). \tag{8}$$

Since the transformed Hamiltonian of Eq. (3) is isotropic, the energy is minimized when the spin expectation values $\left\langle \sum_i \tilde{\mathbf{S}}_i \right\rangle$ and $\left\langle \sum_i \eta_i \tilde{\mathbf{S}}_i \right\rangle$ are parallel to the corresponding effective fields $\mathbf{H}_{\mathrm{eff,u}}$ and $\mathbf{H}_{\mathrm{eff,s}}$, respectively. However, in order to compare with experimental results, it is more convenient to express $\tau$ in terms of the magnitude $H$ and orientation $\theta$ of the original laboratory field $\mathbf{H}$. Evaluating Eq. (6) shows that the $H$ and $\theta$ dependences are separable (see Supplementary Note 2 for detailed derivation). The total bulk magnetic torque is the sum of the uniform, the staggered, and the chiral contributions $\tau_{\mathrm{B}} = \tau_{\mathrm{u}} + \tau_{\mathrm{s}} + \tau_\Phi$ with

$$\frac{\tau_{\mathrm{B}}}{H^2} = \tilde{\chi}_{\mathrm{u}}(H) f_{\mathrm{u}}(\theta) + \tilde{\chi}_{\mathrm{s}}(H) f_{\mathrm{s}}(\theta) + \tilde{\chi}_\Phi(H) f_\Phi(\theta). \tag{9}$$

The field-dependence of each contribution is described by the susceptibilities

$$\tilde{\chi}_x(H) = \tilde{\chi}_{0,x} H^{-\zeta_x}, \tag{10}$$

in terms of the constants $\tilde{\chi}_{0,x}$ and the scaling exponents $\zeta_x$, with $x = \{\mathrm{u, s, \Phi}\}$. Since the effective uniform and staggered fields are orthogonal (see Supplementary Note 1), $\zeta_{\mathrm{u}}$ and $\zeta_{\mathrm{s}}$ are generally different. Constant susceptibility (i.e. $|\mathbf{m}| \propto H$) corresponds to the limit $\zeta \rightarrow 0$. The angle-dependence of the torque results entirely from the anisotropy of the effective fields, and is described by

$$f_x(\theta) = -\frac{\mathrm{d}\mathbf{h}_{\mathrm{eff},x}}{\mathrm{d}\theta} \cdot \frac{\mathbf{h}_{\mathrm{eff},x}}{|\mathbf{h}_{\mathrm{eff},x}|^{\zeta_x}}. \tag{11}$$

where $\mathbf{h}_{\mathrm{eff},x} = \mathbf{H}_{\mathrm{eff},x}/H$. These expressions define the torque up to the constants $\tilde{\chi}_{0,x}$ and the scaling exponents $\zeta_x$.

**Bulk torque response of κ-Cu.** To compare with κ-Cu, we computed $\tau$ for fields rotated in the $ac^\star$-plane, and a variety of possible $\tilde{\chi}_{0,x}$ and $\zeta_x$ using Eq. (9) together with ab initio values for the g-tensors and DM-interactions (see Supplementary Table 1). Coordinates were defined similarly to the experiment of Isono et al.[17], in which $\theta$ is the angle between $\mathbf{H}$ and the long axis of an ET molecule, approximated by the principal axis of $(\mathbb{G}_u \cdot \mathbb{G}_u^{\mathrm{T}})$ close to the $a$-axis (see Fig. 1a).

The experimental response[17] for temperatures $T > T^\star$ is consistent with a material deep in a vanilla QSL or paramagnetic state that corresponds to $\zeta_{\mathrm{u}} = 0$, $\zeta_{\mathrm{s}} = 0$ and $\tilde{\chi}_\Phi = 0$, i.e., the spin susceptibilities are field-independent and there is no chiral response (Fig. 2a). In this case, the uniform contribution $\tau_{\mathrm{u}}$ dominates, and the torque arises primarily from weak anisotropy of the uniform g-tensor $\tilde{\mathbb{G}}_{\mathrm{u}}$, providing a simple $\sin 2(\theta - \theta_0)$ dependence with a fixed $\theta_0 \approx 90°$, as defined in Fig. 2a.

For temperatures $T < T^\star$, the experiments of Isono et al.[17] instead revealed a field dependent torque susceptibility that diverges at low temperatures as $\tau/H^2 \propto H^{-\zeta} \sin 2(\theta - \theta_0)$, with $\zeta = 0.76$–0.83, together with a notable angle shift of $\theta_0(H, T)$. Importantly, we find that these observations cannot be reconciled with divergence of any bulk susceptibility. For example, the case of a diverging uniform susceptibility $\tilde{\chi}_{\mathrm{u}}(H)$ (Fig. 2b, $\zeta_{\mathrm{u}} = 0.8$), then $\tau_{\mathrm{u}}$ would dominate at all temperatures, thus providing a nearly identical angle dependence at all $T$, with no shifting $\theta_0$. This scenario would also imply proximity to a ferromagnetic instability, and therefore appears unlikely given the strong antiferromagnetic interactions[20,28–33].

In this context, divergence of the staggered susceptibility $\tilde{\chi}_{\mathrm{s}}(H)$ appears more likely, which may occur near a critical point between a QSL and a Néel phase. We have previously highlighted this scenario[18] as a possible explanation of the μSR response of κ-Cu. However, we find this scenario is also inconsistent with the experimental torque. Figure 2c, d depicts the torque for an exponent $\zeta_{\mathrm{s}} = 0.8$, for different values of $\tilde{\chi}_{\mathrm{s}}/\tilde{\chi}_{\mathrm{u}}$. For a diverging staggered susceptibility, $\tau_{\mathrm{s}}$ shows a saw-tooth like angle dependence instead of the experimentally observed $\sin 2(\theta - \theta_0)$. This behavior stems from the strong angle dependence of the effective staggered field $|\mathbf{H}_{\mathrm{eff,s}}|$, which contributes to the denominator of Eq. (11). The strongly anisotropic staggered g-tensor $\tilde{\mathbb{G}}_{\mathrm{s}}$ leads to field orientations where $|\mathbf{H}_{\mathrm{eff,s}}|$ vanishes linearly as $|\mathbf{H}_{\mathrm{eff,s}}| \sim (\theta - \theta_c)$. For such orientations, a diverging susceptibility $\mathrm{d}^2 E/\mathrm{d}|\mathbf{H}_{\mathrm{eff,s}}|^2$ implies divergence of $\mathrm{d}\tau/\mathrm{d}\theta \sim \mathrm{d}^2 E/\mathrm{d}\theta^2$, thus giving a saw-tooth torque.

Similar considerations apply to the case of a diverging chiral susceptibility, shown in Fig. 2e, f (with $\mathbf{n}||a$ and $\zeta_\Phi = 0.8$). In this case, the saw-tooth form reflects the vanishing of $\langle \mathcal{H}_{\chi,\mathrm{eff}} \rangle$ for

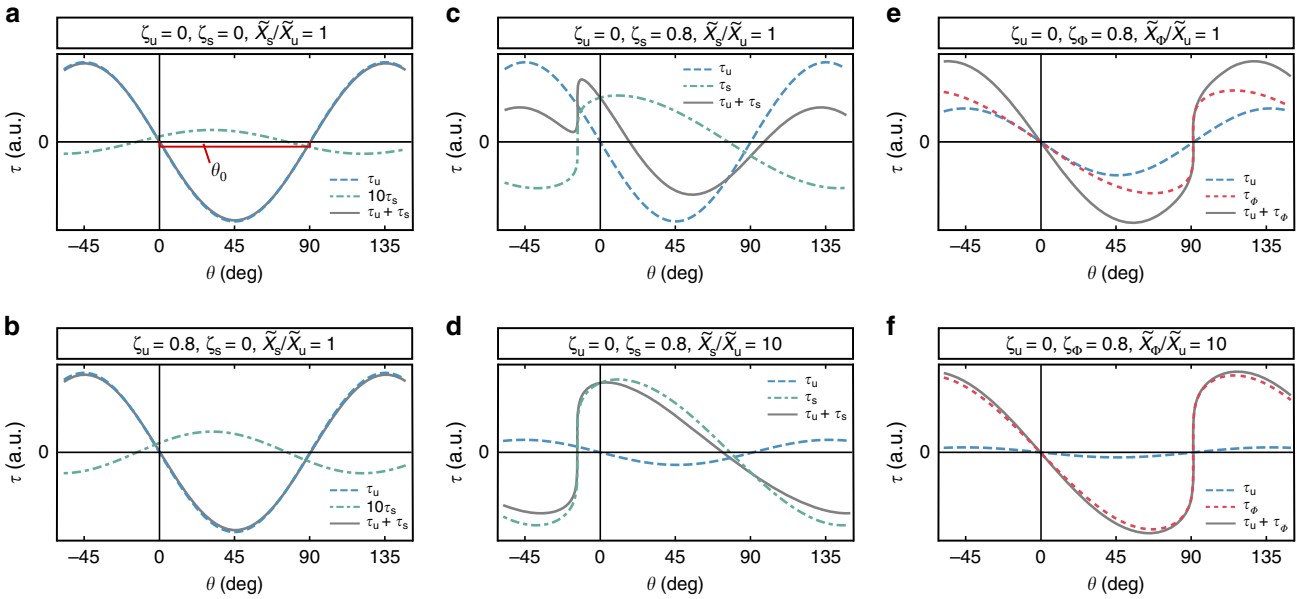

**Fig. 2** Bulk torque angle dependence in critical scenario. Indicated parameter sets distinguish between uniform ($\zeta_u \neq 0$), staggered ($\zeta_s \neq 0$) and chiral ($\zeta_\Phi \neq 0$) criticality, with the susceptibility scaling exponent $\zeta_x$. The total bulk torque is illustrated with solid gray, the uniform with dashed light blue, the staggered with dotted dashed blue, and the chiral contribution with dotted green lines. **a** Conventional case with field-independent susceptibility. **b** Diverging uniform susceptibility ($\zeta_u = 0.8$), dominated only by the uniform contribution. **c** Saw-tooth torque for diverging staggered susceptibility ($\zeta_s = 0.8$) at intermediate field ($\zeta_s = 0.8$) at intermediate field ($\tilde{\chi}_s/\tilde{\chi}_u = 1$). **d** Saw-tooth torque for diverging staggered susceptibility ($\zeta_s = 0.8$) at low-field ($\tilde{\chi}_s/\tilde{\chi}_u = 10$). **e** Saw-tooth torque for diverging scalar chiral susceptibility ($\zeta_\Phi = 0.8$) at intermediate field ($\tilde{\chi}_\Phi/\tilde{\chi}_u = 1$). **f** Saw-tooth torque for diverging scalar chiral susceptibility ($\zeta_\Phi = 0.8$) at low-field ($\tilde{\chi}_\Phi/\tilde{\chi}_u = 10$)

in-plane fields $\mathbf{H} \perp \mathbf{n}$, which leads to similar divergences in $d\tau/d\theta$ for $\zeta_\Phi > 0$. Taken together, the observed field- and temperature dependent angle shift $\theta_0$ and absence of saw-tooth features[17] argue against any diverging bulk susceptibility in $\kappa$-Cu.

**Local spin defects in $\kappa$-Cu.** In order to account for the experimental torque observations at temperatures $T < T^\star$, we considered additional contributions from rare local spin moments induced by disorder, which may also give rise to diverging susceptibilities[34–37]. We identify two relevant types of disorder for $\kappa$-Cu.

The first type produces a random modulation of the magnetic interactions between dimers, and includes disorder in the conformations of the terminal ethylene groups of the ET molecules[38,39], the orientations of the cyanide anions located at crystallographic inversion centers[40], and/or the local charge distributions within each dimer[41–43]. In one-dimensional spin chains, introducing a finite randomness induces a random singlet ground state[44,45], in which the fluctuating singlets of the QSL are randomly pinned to form quasi-static singlets with varying length scales. At any given energy scale, a small fraction of spins remain unpaired, which leads to a diverging susceptibility. As recently emphasized by Kimchi et al.[21,22], similar behavior may occur in higher dimensional analogues of the random singlet state, which we refer to as VBG[46,47] states. A possible scenario[48] occurs in proximity to a valence bond solid order. In this case, random interactions induce complex patterns of domain walls that can host quasi-free orphan spins at their intersections, as depicted in Fig. 3b.

The second type of disorder may result from defects in the anion layer[7,49], which may slightly dope the organic layer, producing nonmagnetic spin vacancies. As illustrated in Fig. 3a, these vacancies break singlet bonds, which may produce local moments if the host system is in a confined state (e.g., a valence bond solid)[50–55]. Interestingly, Furukawa et al.[56] recently showed that the introduction of anion defects via irradiation could

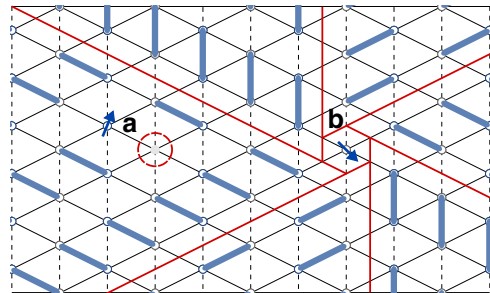

**Fig. 3** Local valence bond defects. Illustrated are two mechanisms for formation of local spin moments on the anisotropic triangular lattice, with singlet bonds indicated in blue and domain walls between valence bond patterns indicated in red. **a** Local spin 1/2 caused by the breaking of a singlet bond due to an anion layer vacancy, emphasized by the red circle. **b** Local spin 1/2 due to a defect in a quasi-static valence bond singlet pattern

suppress magnetic order in the less magnetically frustrated $\kappa$-(ET)$_2$Cu[N(CN)$_2$]Cl salt.

Regardless of their origin, the low-energy response of disorder-induced orphan spins has been the subject of many recent works[23,24,34,36,48,57], including those with specific reference to $\kappa$-Cu[58,59]. In the following, we start from the assumption that such local moments exist at low energies, and are randomly distributed in the material. We assume that the orphan spins interact via random long-range interactions that arise from bulk fluctuations, and consider their contribution to the magnetic torque.

**Defect torque expressions.** For simplicity, we consider the case of $S = 1/2$ impurity spins embedded in a nonordered background, and assume constant bulk susceptibilities $\tilde{\chi}_u$ and $\tilde{\chi}_s$. Due to fluctuations of the valence bonds around each impurity, the

associated magnetic moment will be delocalized over some characteristic localization length. An important distinction from the bulk case is that all Fourier components of the induced defect magnetization density are parallel or antiparallel at lowest order[52]. Therefore, the impurity induced magnetism can be described by a single effective impurity spin variable $\tilde{\mathbf{S}}_I$, which represents both the impurity and the broad screening cloud surrounding the impurity. An external field couples to $\tilde{\mathbf{S}}_I$ through the uniform and staggered moments induced near the impurity. These are given by $\sum_{i' \sim m} \langle \tilde{\mathbf{S}}_{i'} \rangle = c_u \langle \tilde{\mathbf{S}}_{I,m} \rangle$ and $\sum_{i' \sim m} \eta_{i'} \langle \tilde{\mathbf{S}}_{i'} \rangle = c_s \langle \tilde{\mathbf{S}}_{I,m} \rangle$ with $\eta_{i'} = \pm 1$, respectively. Here, the summation runs over dimer sites $i'$ near the impurity $m$, and $c_u$ and $c_s$ are constants related to the Fourier transform of the induced spin density at $k = 0$ and $k = (\pi, \pi)$, respectively. At lowest order, the scalar spin chirality is not coupled to the impurity effects. Therefore only the staggered and uniform moments respond to an external field through

$$\mathcal{H}_{\text{Zee,I}} = -\mu_B \sum_m \mathbf{H} \cdot \tilde{\mathbb{G}}_I \cdot \tilde{\mathbf{S}}_{I,m}, \tag{12}$$

where the effective impurity g-tensor is $\tilde{\mathbb{G}}_I \approx (c_u \tilde{\mathbb{G}}_u + c_s \tilde{\mathbb{G}}_s)$, and therefore differs from any of the bulk g-tensors. The effective field felt by each impurity is given by

$$\mathbf{H}_{\text{eff,I}} = \tilde{\mathbb{G}}_I^T \cdot \mathbf{H}, \tag{13}$$

with the corresponding reduced field $\mathbf{h}_{\text{eff,I}} = \mathbf{H}_{\text{eff,I}}/H$. For simplicity, we take $\tilde{\mathbb{G}}_I$ to be the same for each impurity.

In the absence of residual interactions between $S = 1/2$ impurities, they would act as independent Curie spins, with the impurity contribution to the torque $\tau_I/H^2$ diverging as $T^{-1}$ and $H^{-1}$ at low-field and temperature. However, the overlap of the screening clouds leads to random residual interactions $\{J_{mn}^{\text{eff}}\}$ between the randomly distributed impurity moments $m$ and $n$. Following previous works[23,24,34,36], we expect the interacting impurities to become successively coupled into clusters as the energy is lowered below the interaction energy (e.g., $k_B T \lesssim \max |J_{mn}^{\text{eff}}|$). The initial distribution of interactions may include both ferromagnetic ($J_{mn}^{\text{eff}} < 0$) and antiferromagnetic ($J_{mn}^{\text{eff}} > 0$) terms, as the sign of the coupling depends on the relative positions of the impurities. Provided this distribution is not too singular[24], the low-energy response of the impurities is therefore expected to be described by a finite randomness large-spin fixed point (LSFP)[23,24] or the spin-glass fixed point (SGFP)[36]. In the vicinity of such a fixed point, we find that the total impurity torque can be approximated by a modified Brillouin function $\mathcal{B}(S, x)$ (see Supplementary Note 3)

$$\frac{\tau_I}{H^2} \approx g(\theta) \frac{N_C S_{\text{eff}}^{\text{avg}}}{H} \mathcal{B} \left( S_{\text{eff}}^{\text{avg}}, \frac{\mu_B |\mathbf{H}_{\text{eff,I}}|}{k_B T} \right), \tag{14}$$

$$g(\theta) = -\left( \frac{d\mathbf{h}_{\text{eff,I}}}{d\theta} \cdot \frac{\mathbf{h}_{\text{eff,I}}}{|\mathbf{h}_{\text{eff,I}}|} \right), \tag{15}$$

where $N_C = N_0 \Omega^{2\kappa}$ is the number of clusters, and $S_{\text{eff}}^{\text{avg}} = S_0 \Omega^{-\kappa}$ is the average moment per cluster. Both depend on an effective energy scale defined by

$$\Omega = \max \left( k_B T, \mu_B S_{\text{eff}}^{\text{avg}} |\mathbf{H}_{\text{eff,I}}| \right). \tag{16}$$

The constants $S_0$ and $N_0$ define the average cluster spin and density at fixed $T$ and $H$. The nonuniversal exponent $\kappa$ is related to the fixed point distribution of energy couplings, and is sample-

dependent but constrained by $1 \lesssim (2\kappa)^{-1} \leq \infty$ (see Supplementary Note 3).

Evaluating the torque in the high-field limit $k_B T \ll \mu_B S_{\text{eff}}^{\text{avg}} |\mathbf{H}_{\text{eff,I}}|$, yields that $\tau_I$ is temperature independent

$$\frac{\tau_I}{H^2} \approx \tilde{\chi}_I^H(H) f_I^H(\theta), \tag{17}$$

$$\tilde{\chi}_I^H(H) = \tilde{\chi}_{0,I}^H H^{-\zeta_I}, \tag{18}$$

where $\tilde{\chi}_{0,I}^H$ is a constant that depends on the parameters $\{N_0, S_0, \zeta_I\}$ (see Supplementary Note 3). The angle dependence is given by

$$f_I^H(\theta) = -\frac{d\mathbf{h}_{\text{eff,I}}}{d\theta} \cdot \frac{\mathbf{h}_{\text{eff,I}}}{|\mathbf{h}_{\text{eff,I}}|^{\zeta_I}}. \tag{19}$$

The susceptibility diverges as $\tilde{\chi}_I^H \propto H^{-\zeta_I}$, with the exponent $\zeta_I = (1 + \kappa)^{-1}$ being restricted to the narrow range $2/3 \lesssim \zeta_I \leq 1$ due to the constraints for the nonuniversal exponent $\kappa$. As a result, the low temperature divergence of $\tau_I/H^2$ with respect to field is always marginally weaker than free Curie spins.

In the low-field limit $\mu_B S_{\text{eff}}^{\text{avg}} |\mathbf{H}_{\text{eff,I}}| \ll k_B T$, the torque susceptibility evaluates to

$$\frac{\tau_I}{H^2} \approx \tilde{\chi}_I^T(T) f_I^T(\theta), \tag{20}$$

$$\tilde{\chi}_I^T(T) = \tilde{\chi}_{0,I}^T \left( \frac{1}{k_B T} + S_0^{-1} (k_B T)^{\frac{1}{\zeta_I} - 2} \right), \tag{21}$$

which is independent of field. The derivation of the temperature-independent contribution $\tilde{\chi}_{0,I}^T$ is given in Supplementary Note 3. The angle dependence $f_I^T(\theta)$ is described by

$$f_I^T(\theta) = -\frac{d\mathbf{h}_{\text{eff,I}}}{d\theta} \cdot \mathbf{h}_{\text{eff,I}}. \tag{22}$$

As discussed in the next section, despite not following a perfect power law, the low-field divergence of $\tau_I/H^2$ with respect to temperature may appear to follow a power law $\sim T^{-\omega_I}$ with $2 - \zeta_I^{-1} \leq \omega_I \leq 1$ at intermediate temperatures.

**Total torque response of $\kappa$-Cu.** Here, we show that impurity-related orphan spin contributions can reproduce all observed features of the torque response.

In Fig. 4a, b we show the low temperature angle-dependence of the torque for $\mathbf{H}$ rotated in the $ac^\star$-plane. Including impurity contributions, the total experimental torque is $\tau = \tau_B + \tau_I$ according to Eqs. (9) and (17), respectively. We employed ab initio values (given in Supplementary Table 1), and an impurity exponent $\zeta_I = 0.8$. We assume that the bulk susceptibility is not diverging (i.e., $\zeta_u = \zeta_s = \zeta_\Phi = 0$), and assume the same order of magnitude $\tilde{\chi}_{I,0}^H = \tilde{\chi}_{0,u} = \tilde{\chi}_{0,s}$ for simplicity. In agreement with the measurements for $\kappa$-Cu, the total torque has a sinusoidal $\sin 2(\theta - \theta_0)$ shape. Owing to a nearly isotropic impurity g-tensor $\tilde{\mathbb{G}}_I$, a diverging $\tau_I/H^2$ does not lead to a saw-tooth appearance. However, provided the existence of a finite DM interaction, then $\tilde{\mathbb{G}}_I$ will generally differ from the bulk $\tilde{\mathbb{G}}_u$, leading to a shifted angle dependence of the impurity contribution. This explains the experimentally observed[17] field and temperature dependent angle shift $\theta_0(H,T)$. As illustrated in Fig. 4c, the value of $\theta_0$ measures the ratio of impurity and bulk susceptibilities $\tilde{\chi}_I/\tilde{\chi}_u$. The asymptotic value is further controlled by the ratio of the constants $c_s/c_u$, corresponding to the staggered ($\mathbf{k} = (\pi, \pi)$) and uniform ($\mathbf{k} = 0$) contributions of the induced spin density. For $c_s/c_u = 1$ (Fig. 4a) the impurity and bulk contributions have nearly identical angle dependence, since $\tilde{\mathbb{G}}_I \approx \tilde{\mathbb{G}}_u$. This leads to $\theta_0 \sim -90°$. In contrast, for $c_s/c_u > 1$ (Fig. 4b), the impurity contribution is shifted with respect to the bulk contribution

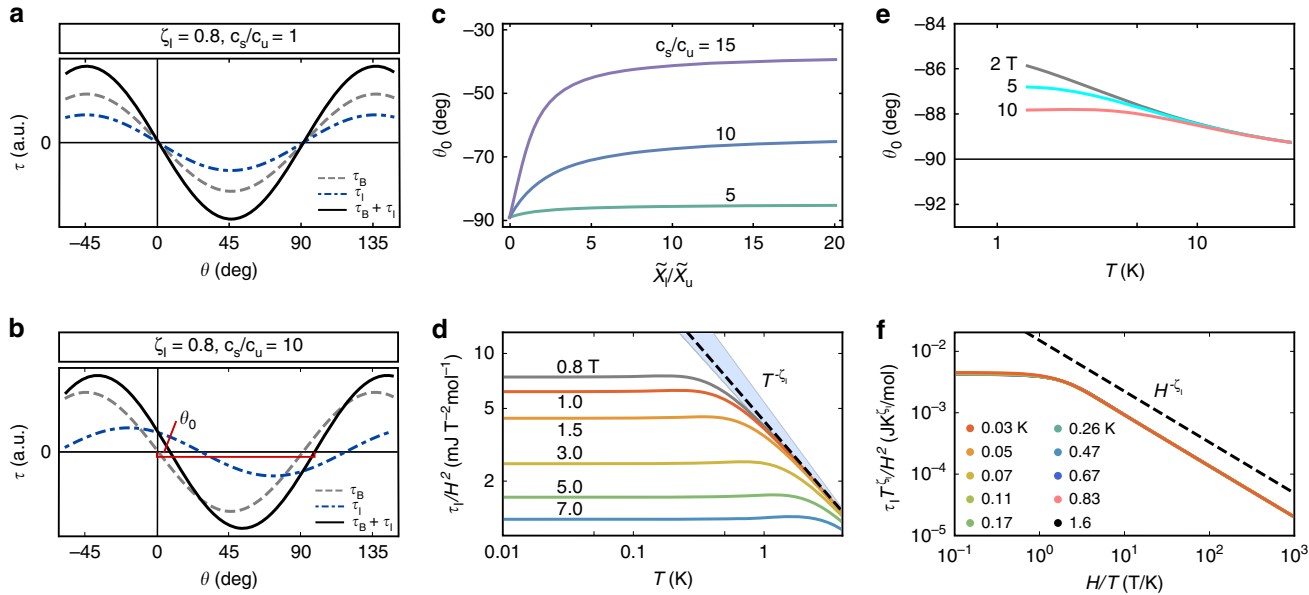

**Fig. 4** Characteristic torque features in defect spin scenario. **a**, **b** Angle-dependence of the magnetic torque $\tau$ for identical impurity and bulk susceptibility contributions ($\tilde{\chi}_I/\tilde{\chi}_u = 1$), conventional bulk susceptibility scaling ($\zeta_B = 0$), and different ratios of induced staggered ($c_s$) and induced uniform ($c_u$) spin density contribution. The total torque is indicated in solid black, the bulk contribution in dashed gray and the impurity contribution in dotted-dashed blue lines. **c** Angle shift $\theta_0$ in terms of $\tilde{\chi}_I/\tilde{\chi}_u$ for indicated ratios $c_s/c_u$. **d** Impurity torque susceptibility $\tau_I/H^2$ as a function of temperature $T$ for various magnetic fields $H$ (compare with ref. [17], Fig. 4d). **e** Angle shift $\theta_0$ as a function of temperature for various magnetic fields (compare with ref. [17], Fig. 2c). **f** Plot of $\tau_I T^{\zeta_I}/H^2$ showing apparent $H/T$ scaling over several orders of magnitude (compare with ref. [17], Fig. 5a). Plots **d–f** employ parameters from fitting the experimental data

(Fig. 4b), which leads to an overall shift of the total torque curve. The experimentally observed angle shift[17] corresponds to $c_s/c_u \approx$ 5–10, implying the staggered moment induced around each defect exceeds the uniform defect moment.

To produce Fig. 4d–f, we performed a global fit of the divergent experimental[17] torque at various $T$, $H$, yielding values for $N_0, S_0$ and $\zeta_I$ in Eq. (14). The fitted exponent ($\zeta_I = 0.79 \pm 0.03$) falls in the middle of the suggested range $2/3 \lesssim \zeta_I \leq 1$, and is therefore consistent with a scenario with orphan spins. Figure 4d shows the resulting temperature dependence at various fields. Consistent with the experiment, the theoretical $\tau_I/H^2 \sim T^{-\omega_I}$ appears to follow a power law behavior. From Eq. (21) it follows that the exponent should fall in the narrow range $2 - \zeta_I^{-1} \leq \omega_I \leq 1$, illustrated by the blue region in Fig. 4d. For this reason, $\omega_I \approx \zeta_I$ (dashed line) at low-field. Interestingly, the similar values of $\zeta_I$ and $\omega_I$ mean that $\tau_I$ will accidentally appear to display $H/T$ scaling described by $\tau_I/H^2 \approx T^{-\zeta_I}F[H/T]$, with

$$F[X] = \begin{cases} \text{constant} & X \ll 1 \\ X^{-\zeta_I} & X \gg 1 \end{cases}. \qquad (23)$$

This behavior is displayed in Fig. 4f, where the theoretical $(\tau_I/H^2)T^{\zeta_I}$ is plotted against $(H/T)$ for different temperatures. Deviations from scaling appear as a very small separation of the curves at low-fields. The apparent data collapse on a general scaling function is due to the disorder induced mechanism discussed in ref. [21]. Finally, in Fig. 4e we show the predicted evolution of the angle shift $\theta_0$, for a fixed ratio of induced staggered to uniform defect moments $c_s/c_u = 6$. At high temperature, $\theta_0$ asymptotically approaches 90°, as thermal fluctuations suppress the diverging impurity contribution.

Taken together, the impurity contributions appear to explain all essential features of the experimental torque response below $T^*$, including the reported range of exponents $\zeta_{exp} = 0.76 - 0.83$, the $\sin 2(\theta - \theta_0)$ dependence, the evolution of the angle shift $\theta_0$, and the apparent $H/T$ scaling of the magnitude of the torque.

**Inhomogeneous NMR response**. Within the same framework, we have also considered the inhomogeneous broadening of the NMR lines in $\kappa$-Cu, which has previously been attributed to impurity spins[4,60]. As the external field aligns the total spin of each cluster, the local defect moments $\tilde{S}_{I,m}$ contributing to the cluster become static on the NMR timescale. The resulting static spin density around each defect is staggered, and decays with distance, leading to an inhomogeneous distribution of staggered Knight shifts within the sample. At first approximation, we assume that the resulting contribution to the NMR linewidth $\nu_I$ scales as the root mean squared average of $\tilde{S}_{I,m}$, leading to (see Supplementary Note 3)

$$\nu_I \approx \nu_0 \mathcal{B}\left(S_{eff}^{avg}, \frac{\mu_B |\mathbf{H}_{eff,I}|}{k_B T}\right). \qquad (24)$$

For the high-field limit $k_B T \ll \mu_B S_{eff}^{avg} |\mathbf{H}_{eff,I}|$, the local impurities are completely static, leading to constant linewidth ($\nu_I = \nu_0$). In the opposite limit $k_B T \gg \mu_B S_{eff}^{avg} |\mathbf{H}_{eff,I}|$, expanding the Brillouin function yields

$$\nu_I \propto \frac{H}{T^{1/\zeta_I}}\left(1 + S_0^{-1}(k_B T)^{\frac{1-\zeta_I}{\zeta_I}}\right). \qquad (25)$$

suggesting $\nu_I$ is linear with respect to field.

In Fig. 5, we show the predicted relative linewidth $\nu_I/\nu_0$ as a function of temperature and field employing $S_0$, $N_0$ and $\zeta_I$ fit from the experimental $\tau$. Comparing Eq. (25) to the experimental behavior, we find reasonable agreement. The experimental linewidth saturates (ref. [4], Fig. 2) in precisely the same temperature range where $\tau/H^2$ also becomes temperature independent (ref. [17], Fig. 3). For $T > 1$ K, the experimental NMR linewidth also grows approximately linear with $H$ (ref. [4], Fig. 4). This correspondence suggests that the NMR linewidth and torque response have a common impurity origin.

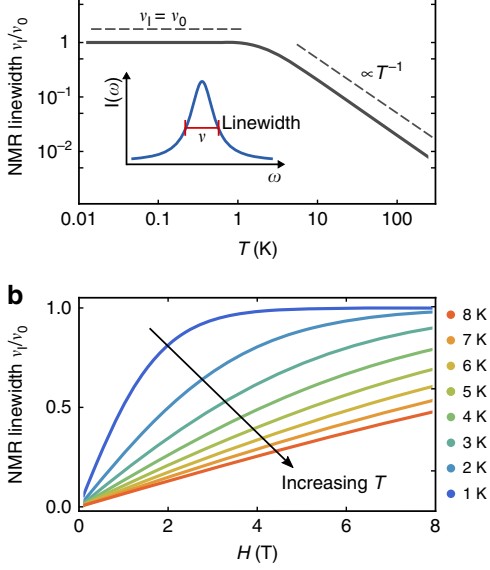

**Fig. 5** Relative impurity contribution to the $^{13}$C NMR linewidth. The relative impurity linewidth $\nu_I/\nu_0$ is calculated according to Eq. (24). **a** $\nu_I/\nu_0$ as a function of temperature $T$ for fixed field $H = 4$ T. Emphasized are the constant linewidth ($\nu_I = \nu_0$) at low temperatures and the $\propto T^{-1}$ scaling at high temperatures. The inset shows the linewidth definition in an NMR spectrum cartoon with the intensity $I$ as a function of frequency $\omega$. **b** $\nu_I/\nu_0$ as a function of field $H$ for various temperatures $T$. The approximately linear field dependence for increasing $T$ is in agreement with experiment[4]

## Discussion

In the presence of anisotropic interactions beyond the conventional Heisenberg terms, measurements of magnetic torque $\tau$ may provide significant insights into quantum magnets. We have shown that the effects of SOC allow the torque to probe uniform and staggered components of spin susceptibilities. In addition, higher order ring-exchange processes, resulting from proximity to a metallic state, directly couple the magnetic field to the scalar spin chirality. For this reason, the torque provides a direct probe for a number of possible instabilities of the spin-liquid state in $\kappa$-Cu, including proximity to magnetically ordered $(\pi, \pi)$ Néel or chiral ordered phases. However, we showed that the experimental response at low temperature is incompatible with the divergence of any bulk susceptibility, arguing against proximity to any such instability.

We therefore considered contributions to $\tau/H^2$ from rare impurity spins. After deriving approximate expressions for such contributions, we find that effects of local orphan spins can explain essentially all features of the torque experiments in $\kappa$-Cu, including the specific angle dependence, observed exponents, and the apparent $H/T$ scaling noted by Isono et al.[17]. Related expressions also consistently describe the inhomogeneous NMR linewidth[4] with the same parameters. These independent experiments both point to the presence of local moments in $\kappa$-Cu, which we attribute to a disorder-induced VBG[46,47] ground state analogous to the random-singlet phase of 1D spin chains[44,45]. While the small fraction of orphan spins result in a diverging torque response, the majority of low-energy excitations will consist of local domain wall fluctuations, i.e., shifts of the frozen valence bond pattern[47]. Such fluctuations can give rise to a linear specific heat[58,61] $C_\nu \sim T$, in analogy with structural glasses[62]. Since most excitations of the VBG are ultimately localized, they will contribute negligibly to thermal conductivity $\kappa_T$ in the $T \to 0$ limit. This feature may therefore explain the linear $C_\nu$ but suppressed $\kappa_T$ observed in $\kappa$-Cu[63,64]. Since disorder is expected to be small compared to the

scale of interactions, the appearance of local moments also places some constraints on the ground state of the hypothetical disorder-free sample[50–55]. In particular, proximity to valence bond solid order may aid the formation of a VBG[48].

We note, however, that an important discrepancy between the scaling expressions and the experimental response occurs at the $T^\star = 6$ K anomaly. At $T^\star$, both the experimental torque angle shift $\theta_0$ and the NMR linewidth $\nu$ increase more sharply than suggested by the scaling ansätze, so that strong contributions from orphan spins seem to emerge rapidly at $T^\star$. We, therefore, speculate that the anomalies observed in a wide range of experiments are connected to the freezing of valence bonds into the random VBG configuration. In principle, this may be driven by a number of effects. For example, it may reflect a thermal valence bond solid ordering transition in hypothetical disorder free samples, which evolves into a VBG transition with finite disorder. Alternately, external factors such as freezing of local charges or anion orientations may lead to a rapid enhancement of the effective disorder, driving the rapid formation of the VBG. In either case, the singlets would be free to fluctuate for $T \gg T^\star$, thus forming an essentially homogeneous QSL at high temperatures. Strong inhomogeneity in the NMR relaxation (and $\mu$SR response[6]) would only be expected below $T^\star$, which is consistent with ref. [4]. Moreover, the temperature $T^\star$ itself should not be strongly sensitive to external fields, which is consistent with negligible field dependence[9] of the thermal expansion near $T^\star$.

On this basis, we conclude that the torque investigations of $\kappa$-Cu provide unique insight into the low temperature phase. The analysis presented in this work provide direct evidence of disorder-related effects and may provide a roadmap for finally revealing the nature of the enigmatic $T^\star$ anomaly as the formation of a VBG. This framework may also be generalized to other prominent organic QSL candidates like $\kappa$-H$_3$(Cat-EDT-TTF)$_2$[65] or $\beta'$-EtMe$_3$Sb[Pd(dmit)$_2$]$_2$[66]. It is also noteworthy that similar scaling of the susceptibility with exponent $\zeta = 2/3$ is observed in the frustrated Kagome system Herbertsmithite[67], where disorder is thought to play a role. This highlights the rich interplay between disorder and frustration in quantum spin materials.

### Data availability
Data is available from the corresponding author upon reasonable request.

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

## Acknowledgements

The authors acknowledge fruitful discussions with S. Brown, V. Dobrosavljevic, S. Hartmann, K. Kanoda, I. Kimchi, M. Lang, and P. Szirmai. The work was supported by the Deutsche Forschungsgemeinschaft (DFG) through project SFB/TRR49.

## Author contributions

All authors contributed to all aspects of this work.

## Additional information

**Competing interests:** The authors declare no competing interests.

