## [Peer Review File · Nature Communications]

Reviewers' comments:

Reviewer #1 (Remarks to the Author):

In this manuscript, the authors theoretically discuss the magnetic properties observed in a triangular lattice, k-ET salt. Some organic salts with a $S=1/2$ triangular lattice have been extensively studied because of the fascinating magnetic properties, quantum spin liquid (QSL) behavior. Among them, the spin state of k-ET salt is very unique in that the magnetic torque shows quantum critical behavior below the well-known 6 K anomaly. This theoretical model explains the critical behavior in terms of disorder-induced spin defects and the 6 K anomaly is ascribed to a random valence bond glass. The heat capacity proportional to temperature is interpreted by the local domain wall fluctuations. This model is very interesting and may be consistent with the experimental results. However, I have a very basic question on this theoretical model.

In addition to k-ET salt, there are two other QSL materials, β' -(Cation)[Pd(dmit)₂]₂ and k-H3(Cat-EDT-TTF)₂, which do not show such random valence bond glass. What are different between them?

Reviewer #2 (Remarks to the Author):

This is a theoretical paper, devoted to the low-temperature behavior of the organic Mott insulator κ -(ET)₂-Cu₂(CN)₃. This material is a candidate for a quantum spin liquid, with magnetic spin-1/2 moments forming an effective triangular lattice. Recent magnetic torque measurements have detected unconventional behavior, which was tentatively assigned to quantum criticality. The present authors argue that the behavior instead originates from defect physics, and use a phenomenological model based on a temperature-dependent distribution of free moments to fit experimental data.

After having studied the paper, I think that the work is in principle important enough to quality for publication in Nature Communications because a proper interpretation of experiments performed on spin-liquid candidate materials is crucial for progress in the field of frustrated magnetism. I feel that the arguments given by the authors are reasonable, and I am willing to believe that their theory (which heavily draws from previous work on disordered spin systems) is closer to the truth than the originally proposed quantum critical one. The agreement with experimental data is impressive, although a few fit parameters are involved. However, I also feel that the story is incomplete, and important questions remain unanswered. Let me be precise:

(i) The authors assume (but not derive) the existence of quasi-free defect spins. This is a highly non-trivial assumption. While it is established that defects in dimer spin systems or in valence-bond solids generate local moments (Fig 3 of the paper), this is **not** obviously true in a deconfined spin liquid: Deconfinement here implies that a vacancy (which may be thought of as liberating a spinon in a RVB-like picture) does **not** bind this spinon. The fact that a vacancy does **not** induce a local moment has also been verified numerically for the kagome Heisenberg model (e.g. PRB 68, 224416 (2003)). Similarly, it is not obvious that weak bond disorder in a deconfined spin liquid produces quasi-free moments. Hence, the microscopic mechanism which would lead to the assumed defect spins is not clear (unless bond disorder is strong). Invoking the physics depicted in Fig 3 would rely on close proximity to a valence-bond-solid phase, but this is not discussed by the authors.

(ii) The role and nature of the anomaly at 6 K is still unclear. According to the authors, disorder effects become strong below this temperature scale, but the reason is open. The authors tentatively assign the anomaly with a glass transition into a valence-bond glass, but it is not clear to this referee whether this is a speculation or whether independent experimental evidence exists.

Given that 6 K is small compared to the exchange scale, this may point to weak disorder, but then leaves open how defect spins emerge in the first place, see point 1.

(iii) Thermal conductivity measurements in Ref 63 have deduced a small energy gap of 0.5 K - is this consistent with the authors' phenomenology?

In summary, this is a high-level paper with results relevant for the community working on frustrated magnets. However, the present version leaves many open questions, and a more in-depth discussion of the issues listed above is required before the paper can be considered for publication.

Reviewer #3 (Remarks to the Author):

In this theoretical work, the authors claim that "we show that disorder-induced spin defects provide a comprehensive explanation of the low-temperature properties" for the triangular-lattice antiferromagnet κ -(ET)₂Cu₂(CN)₃".

The authors focus on torque, and also discuss NMR. Experimentally evidently the torque shows particular divergences at low temperatures and fields. These divergences are similar to what one finds for isolated anisotropic free spins, but modified. The authors argue that the modification can be achieved by considering interactions between disorder induced spin defects.

In particular for torque they claim to reproduce:

- reported range of exponents $\zeta_{\text{exp}} = 0.76 - 0.83$,
- the $\sin 2(\theta - \theta_0)$ dependence,
- the evolution of the angle shift θ_0 ,
- and the apparent H/T scaling of the magnitude of the torque

I understand their argument for the first three items. Appendix 3 is crucial for item 1, but that's fine. However I cannot follow the argument for the last item. It is phrased as follows

"In theory,

the apparent exponent should fall in the narrow range

$$2 - \zeta$$

$$- 1$$

$I \leq \omega I \leq 1$, illustrated by the blue region in Fig. 4d.

For this reason, $\omega I \approx \zeta I$ (dashed line) at low-field. Interestingly, the similar values of ζI and ωI mean that τI will

accidentally appear to display H/T scaling described by

$$\tau I / H^2 \approx T$$

$$= \zeta I [H/T]"$$

Where does the first inequality " $2 - \zeta^{-1} \leq \omega I \leq 1$ " come from?

The NMR discussion is fine.

In terms of impact, I feel that this paper is sufficient to warrant publication in Nat Comm, assuming the technical point above can be clarified. The paper is technical and opaque to read. But it suggests an interesting and consistent theoretical explanation of unusual experimental data, so it is hard to avoid the technical discussions.

Making it easier to read by reminding the reader of definitions of variables would go a long way

towards readability. For example what are the exact definitions of the lowercase c constants `c_s`, `c_u`?

I. SUMMARY OF CHANGES

Main text:

- We revised the manuscript with several additional comments and callouts to improve the readability of the manuscript per suggestion of referee # 3.
- We have improved the discussion of the origin and consequences of the local moments in sections entitled “Local spin defects in κ -Cu” and “Discussion”.

Supplementary Material:

- We slightly modified the derivations in the supplemental to match the updated notation now used in the main text.

II. RESPONSE TO REFEREE # 1

Referee # 1: *In this manuscript, the authors theoretically discuss the magnetic properties observed in a triangular lattice, k -ET salt. Some organic salts with a $S=1/2$ triangular lattice have been extensively studied because of the fascinating magnetic properties, quantum spin liquid (QSL) behavior. Among them, the spin state of k -ET salt is very unique in that the magnetic torque shows quantum critical behavior below the well-known 6 K anomaly. This theoretical model explains the critical behavior in terms of disorder-induced spin defects and the 6 K anomaly is ascribed to a random valence bond glass. The heat capacity proportional to temperature is interpreted by the local domain wall fluctuations. This model is very interesting and may be consistent with the experimental results.*

Authors' Response: We thank the referee for their positive assessment of our work.

Referee # 1: *However, I have a very basic question on this theoretical model. In addition to k -ET salt, there are two other QSL materials, β' -(Cation)[Pd(dmit)₂]₂ and k -H₃(Cat-EDT-TTF)₂, which do not show such random valence bond glass. What are different between them?*

Authors' Response: We thank the referee for raising this point. Similar to κ -(ET)₂Cu₂(CN)₃, both compounds mentioned by the referee can be mapped onto an anisotropic lattice with effective $S = 1/2$. In the following we discuss the two compounds mentioned by the referee.

- β' -EtMe₃Sb[Pd(dmit)₂]₂: Watanabe *et al.* reported the magnetic torque measurements on β' -EtMe₃Sb[Pd(dmit)₂]₂ in Nat. Commun. **3**, 1090 (2012). In this work, the authors look for quantum oscillations of the torque as a possible signature of a spinon Fermi surface, and therefore they have not focused on the experimental aspects relevant to impurity contributions.

[redacted]

[redacted]

An important difference of this material to κ -(ET)₂Cu₂(CN)₃ is the stacking motif of the organic molecules, shown in the left panel of the figure above. For β' -EtMe₃Sb[Pd(dmit)₂]₂, all dimers ($S = 1/2$ sites) are related by inversion centers. Hence, a finite nearest-neighbour Dzyaloshinskii-Moriya interaction and a staggered pattern of the g -tensor are forbidden. In our approach to analyzing the magnetic torque contributions due to impurities, the absence of a DM-interaction and staggered g -tensor in the bulk implies that the effective g -tensor of each impurity is essentially equal to the bulk g -tensor. As a result, a diverging response from the impurities would not lead to an angle shift $\theta_0(H)$ for β' -EtMe₃Sb[Pd(dmit)₂]₂, which makes detection of impurity contributions somewhat more difficult. To our knowledge, no angle shift has been observed in the experiments of Watanabe *et al.*.

[redacted]

The apparent linear field dependence (i.e. $\tau \propto H^2$) is a signature of conventional magnetic behaviour corresponding to an exponent $\zeta \rightarrow 0$. There appears to be no diverging contributions to the torque response at low fields that could be attributed to orphan spins or criticality. It may be useful to reinvestigate the low-field torque to confirm the absence of a diverging contribution before drawing any strong conclusions.

- κ -H₃(Cat-EDT-TTF)₂: Magnetic torque measurements on κ -H₃(Cat-EDT-TTF)₂ were reported by Isono *et al.* in Phys. Rev. Lett. **112**, 177201 (2014).

[redacted]

[redacted]

The organic dimers in $\kappa\text{-H}_3(\text{Cat-EDT-TTF})_2$ have the same stacking motif as $\kappa\text{-(ET)}_2\text{Cu}_2(\text{CN})_3$, and hence a temperature and field dependent angle-shift $\theta_0(H, T)$ is expected for this material if impurity contributions arise. Indeed, the experiments observe such a shift, as shown in Fig. 3(c,d), PRL **112**, 177201 (2014). Here, the authors only show data for selected temperatures. However, the angle shift appears to grow large between 4 K and 15 K. It is worth noting that specific heat measurements on $\kappa\text{-H}_3(\text{Cat-EDT-TTF})_2$ show a broad anomaly at ~ 6 K, which is similar to the T^* anomaly in $\kappa\text{-(ET)}_2\text{Cu}_2(\text{CN})_3$ (see Yamashita *et al.*, PRB **95**, 184425 (2017)). Therefore, we might expect a similar phenomenology of the magnetic torque.

Isono *et al.* report that the amplitude of the torque follow essentially a conventional $\tau \propto H^2$ dependence (Fig. 3(e)). However, this is only unambiguously true for the high field regime. [redacted] Without more detailed experimental results, it is difficult for us to draw strong conclusions. [redacted]

III. RESPONSE TO REFEREE # 2

Referee # 2: *This is a theoretical paper, devoted to the low-temperature behavior of the organic Mott insulator $\kappa\text{-(ET)}_2\text{-Cu}_2\text{(CN)}_3$. This material is a candidate for a quantum spin liquid, with magnetic spin-1/2 moments forming an effective triangular lattice. Recent magnetic torque measurements have detected unconventional behavior, which was tentatively assigned to quantum criticality. The present authors argue that the behavior instead originates from defect physics, and use a phenomenological model based on a temperature-dependent distribution of free moments to fit experimental data.*

After having studied the paper, I think that the work is in principle important enough to quality for publication in Nature Communications because a proper interpretation of experiments performed on spin-liquid candidate materials is crucial for progress in the field of frustrated magnetism. I feel that the arguments given by the authors are reasonable, and I am willing to believe that their theory (which heavily draws from previous work on disordered spin systems) is closer to the truth than the originally proposed quantum critical one. The agreement with experimental data is impressive, although a few fit parameters are involved.

Authors' Response: We thank the referee for their assessment of our manuscript and are strongly encouraged by their positive comments.

Referee # 2: *However, I also feel that the story is incomplete, and important questions remain unanswered. Let me be precise:*

*(i) The authors assume (but not derive) the existence of quasi-free defect spins. This is a highly non-trivial assumption. While it is established that defects in dimer spin systems or in valence-bond solids generate local moments (Fig 3 of the paper), this is **not** obviously true in a deconfined spin liquid: Deconfinement here implies that a vacancy (which may be thought of as liberating a spinon in a RVB-like picture) does **not** bind this spinon. The fact that a vacancy does **not** induce a local moment has also been verified numerically for the kagome Heisenberg model (e.g. PRB 68, 224416 (2003)). Similarly, it is not obvious that weak bond disorder in a deconfined spin liquid produces quasi-free moments. Hence, the microscopic mechanism which would lead to the assumed defect spins is not clear (unless bond disorder is strong). Invoking the physics depicted in Fig 3 would rely on close proximity*

to a valence-bond-solid phase, but this is not discussed by the authors.

Authors' Response: We thank the referee for raising this very good point. We completely agree, that the existence of local moments is a non-trivial assumption. This assumption was motivated, in part, by the previous experimental NMR observations suggesting inhomogeneity and local moment formation at low temperatures. Our motivation was to see if both the NMR and unusual torque response could be understood within a common framework. The fact that two independent experiments are consistent with disorder-induced local moments strengthens the case for their presence.

Although we have speculated on the origin of the local moments, the specific mechanism relevant to κ -(ET)₂Cu₂(CN)₃ remains a subject of future investigation. It is likely that there are multiple forms of disorder present in the material, and their relative role may provide clues toward the nature of the ground state.

We agree with the referee that the presence of local moments (below T^*) rules out a scenario where disorder takes the form of nonmagnetic vacancies and the bulk material forms a deconfined QSL. On the other hand, introducing non-magnetic impurities into a valence bond solid may induce local moments.

For the case of random bond disorder, we believe the situation is less clear. For the one dimensional Heisenberg model (nominally a deconfined QSL), it is generally thought that any finite amount of bond randomness leads to an inhomogeneous phase referred to as a random singlet (RS) or valence bond glass (VBG) state. We note that we are using both of these terms interchangeably. In such a state, typical spin-spin correlations are short-ranged, and the system exhibits a diverging susceptibility at low temperature due to the formation of quasi-local moments (i.e. localized spinons). Whether this behaviour extends to higher dimensions is an open subject of investigation. Recently, it has been argued that this behavior is realised in higher dimensional systems - at least, in proximity to a VBS phase (where any finite randomness induces a random singlet phase, see e.g. Phys. Rev. X **8**, 041040 (2018)).

The observation that only some states may host local moments is the essential content of our proposal regarding the T^* anomaly, discussed in more detail below. We agree with the referee that this point needed to be clarified, and we have modified the manuscript accordingly.

Referee # 2: *(ii) The role and nature of the anomaly at 6 K is still unclear. According to the authors, disorder effects become strong below this temperature scale, but the reason is open. The authors tentatively assign the anomaly with a glass transition into a valence-bond glass, but it is not clear to this referee whether this is a speculation or whether independent experimental evidence exists. Given that 6 K is small compared to the exchange scale, this may point to weak disorder, but then leaves open how defect spins emerge in the first place, see point 1.*

Authors' Response: Our speculation is meant to guide future efforts to understand the T^* anomaly, and is based on the following observations. The experimental NMR and torque provide evidence for local moments below T^* . This suggests that the ground state is essentially a random singlet / valence bond glass state. However, the torque and NMR responses that can be directly attributed to local moments seem to appear strong only below T^* . Our purpose is to highlight physical situations consistent with this observation.

We have in mind two scenarios, in principle, that can lead to this behavior. The first scenario is that T^* reflects a thermal confinement-deconfinement transition that exists in the disorder-free limit of the underlying spin model. In this case, the ground state of the disorder-free model would be a VBS, which becomes a random singlet state in the presence of finite disorder. As the temperature is raised through T^* , the singlet bond configurations melt, and the local moments are suppressed, which would explain the absence of local moment contributions above T^* . In a second scenario, the disorder-free spin model may exhibit a deconfined QSL ground state, but disorder is sufficiently strong to localize spinons, inducing a random singlet ground state instead. In the absence of other effects, this case would feature a thermally driven crossover. For temperatures small compared to the disorder strength, the response would be essentially inhomogeneous, and contributions from local moments would be apparent. For temperatures large compared to the disorder strength, the response would be indistinguishable from the deconfined QSL of the disorder-free model. On the basis of our analysis, we cannot distinguish such scenarios. We further note that both above scenarios may be affected by coupling to degrees of freedom outside the spin model. In κ -(ET)₂Cu₂(CN)₃, there has been much speculation regarding the effects of internal charge degrees of freedom within the dimers, which may freeze into random configurations at low

temperatures. Such a freezing would lead to a rapid enhancement of the disorder strength in the effective spin model, which may drive localization.

The common feature of these scenarios is that the state below T^* is a random singlet or valence bond glass phase, while the state above T^* does not possess a high density of local moments. This is the essential point of our speculation.

Referee # 2: *(iii) Thermal conductivity measurements in Ref 63 have deduced a small energy gap of 0.5 K - is this consistent with the authors' phenomenology?*

Authors' Response: The observation of an energy gap with thermal conductivity is indeed consistent with our phenomenology. A puzzle yet to be solved regarding κ -(ET)₂Cu₂(CN)₃ was raised when specific heat measurements (S. Yamashita *et al.*, Nat. Phys. **4**, 459 (2008)) and thermal conductivity measurements (M. Yamashita *et al.*, Nat. Phys. **5**, 44 (2009)) gave seemingly contradicting results. While the linear T dependence of the specific heat was attributed to gapless excitations, the thermal conductivity κ/T vanishes for $T \rightarrow 0$, indicating the gapless excitations are localized.

The scenario of a valence bond glass ground state offers a consistent picture for both experiments. Note that the orphan spins constitute a very small fraction of the sample, and therefore do not contribute strongly to either specific heat or thermal transport at experimental temperatures. In principle, the local moments would give a small divergent contribution to C_V at low temperatures; however, we have estimated this contribution and find that it is likely to be masked by nuclear Schottky contributions that also appear in experiments. With this said, there remain many local degrees of freedom in the valence bond glass state corresponding to fluctuations of domain walls, which may provide a linear T specific heat (per the analogy with structural glasses discussed in Phys. Rev. Lett. **104**, 177203 (2010)). Due to the localized nature of these excitations, they do not contribute to the transport, which could explain the downturn in the thermal conductivity at low temperatures. While these statements are strictly speculations, the scenario of an inhomogeneous ground state may offer a consistent explanation for the seemingly contradicting experiments.

Referee # 2: *In summary, this is a high-level paper with results relevant for the community working on frustrated magnets. However, the present version leaves many open*

questions, and a more in-depth discussion of the issues listed above is required before the paper can be considered for publication.

Authors' Response: We are grateful for the positive assessment of our work and hope that the revised version of the manuscript satisfies the referee's concerns.

IV. RESPONSE TO REFEREE # 3

Referee # 3: *In this theoretical work, the authors claim that “we show that disorder-induced spin defects provide a comprehensive explanation of the low-temperature properties” for the triangular-lattice antiferromagnet κ -(ET) $_2$ Cu $_2$ (CN) $_3$ ”.*

The authors focus on torque, and also discuss NMR. Experimentally evidently the torque shows particular divergences at low temperatures and fields. These divergences are similar to what one finds for isolated anisotropic free spins, but modified. The authors argue that the modification can be achieved by considering interactions between disorder induced spin defects.

In particular for torque they claim to reproduce:

- reported range of exponents $\zeta_{exp} = 0.76 - 0.83$,
- the $\sin 2(\theta - \theta_0)$ dependence,
- the evolution of the angle shift θ_0 ,
- and the apparent H/T scaling of the magnitude of the torque

I understand their argument for the first three items. Appendix 3 is crucial for item 1, but that’s fine.

Authors’ Response: We thank the referee for their clear summary of our manuscript and their positive assessment.

Referee # 3: *However I cannot follow the argument for the last item. It is phrased as follows*

“In theory, the apparent exponent should fall in the narrow range $2 - \zeta_1^{-1} \leq \omega_1 \leq 1$, illustrated by the blue region in Fig. 4d. For this reason, $\omega_1 \approx \zeta_1$ (dashed line) at low-field. Interestingly, the similar values of ζ_1 and ω_1 mean that τ_1 will accidentally appear to display H/T scaling described by $\tau_1/H^2 \approx T^{-\zeta_1} F[H/T]$ ”

Where does the first inequality “ $2 - \zeta_1^{-1} \leq \omega_1 \leq 1$ ” come from?

Authors’ Response: The mentioned inequality follows from the torque susceptibility

in the high-temperature limit, given by Eq. (20) in the manuscript:

$$\tilde{\chi}_I^T(T) = \tilde{\chi}_{0,I}^T \left(\frac{1}{k_B T} + S_0^{-1} (k_B T)^{\frac{1}{\zeta_I} - 2} \right),$$

The goal of the inequality is to identify the temperature scaling $\tilde{\chi}_I^T \propto T^{-\omega_I}$. For intermediate temperatures, i.e. small temperatures for which Eq. (20) of the manuscript still holds, the first term dominates the temperature dependence and we can identify $\omega_I \approx 1$. In the limit of very high temperatures the second term is dominant, implying $\omega_I \approx 2 - \zeta_I^{-1}$. Hence, for intermediate temperatures the apparent temperature exponent fulfills the inequality: $2 - \zeta_I^{-1} \leq \omega_I \leq 1$.

We apologize for the confusing arrangement in the manuscript and added a callout of the relevant equation in the appropriate paragraph in the manuscript.

Referee # 3: *The NMR discussion is fine.*

Authors' Response: We thank the referee for their positive comment.

Referee # 3: *In terms of impact, I feel that this paper is sufficient to warrant publication in Nat Comm, assuming the technical point above can be clarified. The paper is technical and opaque to read. But it suggests an interesting and consistent theoretical explanation of unusual experimental data, so it is hard to avoid the technical discussions.*

Authors' Response: We thank the referee for their understanding regarding the difficulties to avoid a technical discussion.

Referee # 3: *Making it easier to read by reminding the reader of definitions of variables would go a long way towards readability. For example what are the exact definitions of the lowercase c constants c_s, c_u ?*

Authors' Response: We have made an attempt to clarify the notation. We hope that the referee finds the revised version of the manuscript more suitable.

REVIEWERS' COMMENTS:

Reviewer #1 (Remarks to the Author):

The authors appropriately replied to my comments.
The manuscript seems worthwhile publishing.

Reviewer #2 (Remarks to the Author):

The authors' response clarifies most of the issues raised by this and the other reviewers. Although the changes made to the paper are minor, I feel that they have improved the presentation and enhanced the readability of the paper. I maintain that the work is important enough to qualify for publication in Nature Communications. In my previous report I noted that the story is incomplete. While this is still true, I think the authors have done the best they can (at this stage) in interpreting data in this notoriously difficult field. Hence, I recommend publication of the paper.

Reviewer #3 (Remarks to the Author):

The authors have responded convincingly to my earlier questions and the new manuscript looks fine. I also feel that they have responded convincingly to the (quite positive!) reports of the other referees. The paper is technically sound and its impact is significant and suitable for Nat Comm.

One point concerning the authors' summary of the previous literature: it seems that Ref 22, which the authors cite in the context of the appearance of spin defects, claims to derive H/T scaling of susceptibility which looks similar to the H/T scaling found by the authors in their Figure 4f. Is there a reason the analogous H/T collapse described in Ref 22 is not mentioned in this context?

The authors should have the opportunity to consider making a small revision related to this point, as well as any other minor changes related to the other referees. Following this I recommend rapid publication.

I. RESPONSE TO REFEREE # 1

Referee # 1: *The authors appropriately replied to my comments. The manuscript seems worthwhile publishing.*

Authors' Response: We are grateful for the positive assessment of our reply and work.

II. RESPONSE TO REFEREE # 2

Referee # 2: *The authors' response clarifies most of the issues raised by this and the other reviewers. Although the changes made to the paper are minor, I feel that they have improved the presentation and enhanced the readability of the paper. I maintain that the work is important enough to qualify for publication in Nature Communications. In my previous report I noted that the story is incomplete. While this is still true, I think the authors have done the best they can (at this stage) in interpreting data in this notoriously difficult field. Hence, I recommend publication of the paper.*

Authors' Response: We thank the referee for their positive comments about the changes to the manuscript, their understanding regarding the difficulty of the field and their recommendation for publication.

III. RESPONSE TO REFEREE # 3

Referee # 3: *The authors have responded convincingly to my earlier questions and the new manuscript looks fine. I also feel that they have responded convincingly to the (quite positive!) reports of the other referees. The paper is technically sound and its impact is significant and suitable for Nat Comm.*

Authors' Response: We thank the referee for their positive assessment of our revised manuscript.

Referee # 3: *One point concerning the authors' summary of the previous literature: it seems that Ref 22, which the authors cite in the context of the appearance of spin defects, claims to derive H/T scaling of susceptibility which looks similar to the H/T scaling found by the authors in their Figure 4f. Is there a reason the analogous H/T collapse described in Ref 22 is not mentioned in this context?*

Authors' Response: We thank the referee for raising this point. We completely agree with the referee's suggestion and added a comment to the manuscript accordingly:

“The apparent data collapse on a general scaling function is due to the disorder induced mechanism discussed in Ref. 22.”

Referee # 3: *The authors should have the opportunity to consider making a small revision related to this point, as well as any other minor changes related to the other referees. Following this I recommend rapid publication.*

Authors' Response: We thank the referee for their recommendation and hope the revised manuscript satisfies the referee's concerns.